# Impact of COVID-19 Vaccination Rates and Public Measures on Case Rates at the Provincial Level, Thailand, 2021: Spatial Panel Model Analyses

**DOI:** 10.3390/tropicalmed8060311

**Published:** 2023-06-06

**Authors:** Charuttaporn Jitpeera, Suphanat Wongsanuphat, Panithee Thammawijaya, Chaninan Sonthichai, Sopon Iamsirithaworn, Scott J. N. McNabb

**Affiliations:** 1Division of Epidemiology, Department of Disease Control, Ministry of Public Health, Nonthaburi 11000, Thailand; suphanat.wong@gmail.com (S.W.); viewfetp@gmail.com (P.T.); 2Vaccine Preventable Diseases Unit, Division of Communicable Diseases, Department of Disease Control, Ministry of Public Health, Nonthaburi 11000, Thailand; chaninan33@yahoo.com; 3Deputy Director General of the Department of Disease Control, Ministry of Public Health, Nonthaburi 11000, Thailand; iamsiri@gmail.com; 4Hubert Department of Global Health, Rollins School of Public Health, Emory University, Atlanta, GA 30329, USA; scottjnmcnabb@emory.edu

**Keywords:** COVID-19 vaccination, spatial analysis, Thailand

## Abstract

The coronavirus disease of 2019 (COVID-19) was a pandemic that caused high morbidity and mortality worldwide. The COVID-19 vaccine was expected to be a game-changer for the pandemic. This study aimed to describe the characteristics of COVID-19 cases and vaccination in Thailand during 2021. An association between vaccination and case rates was estimated with potential confounders at ecological levels (color zones, curfews set by provincial authorities, tourism, and migrant movements) considering time lags at two, four, six, and eight weeks after vaccination. A spatial panel model for bivariate data was used to explore the relationship between case rates and each variable and included only a two-week lag after vaccination for each variable in the multivariate analyses. In 2021, Thailand had 1,965,023 cumulative cases and 45,788,315 total administered first vaccination doses (63.60%). High cases and vaccination rates were found among 31–45-year-olds. Vaccination rates had a slightly positive association with case rates due to the allocation of hot-spot pandemic areas in the early period. The proportion of migrants and color zones measured had positive associations with case rates at the provincial level. The proportion of tourists had a negative association. Vaccinations should be provided to migrants, and collaboration between tourism and public health should prepare for the new era of tourism.

## 1. Introduction

The SARS-CoV-2 virus causes the coronavirus disease of 2019 (COVID-19); the first outbreak occurred in China in late December 2019 and then spread worldwide [1]. In addition, the health system in many countries, especially in developing countries, collapsed due to overwhelming cases and diagnostic laboratory challenges [2]. The pandemic caused a high number of cases and deaths; as of 1 January 2022, there had been 288 million cases and 4.8 million deaths globally [3]. Currently, there is no specific treatment for COVID-19. Thus, all countries were trying to stop the pandemic’s spread using public health measures (e.g., home quarantine, keeping social distance, travel restrictions, shutting down cities). However, these measures also had negative consequences by affecting each country’s economy and the population’s mental health. Under the state of emergency, COVID-19 vaccines had been invented, approved, and implemented to control the pandemic since December 2020 and aimed to be a game-changer for the pandemic [4]. Each country’s policymakers decided to purchase and prioritize a target population for vaccination.

Thailand was the first country to detect the first COVID-19 case outside of China, where the majority of cases were imported and local transmission was discovered in early 2020 [5,6]. In 2021, Thailand confronted three waves of COVID-19 that were titled based on the COVID-19 variants: the Second Wave from B.1.36.16 or Bangladesh/UK lineage (18 December 2020 to April 2021), the Third Wave from Alpha or B.1.1.7 variant (2 April to 5 July 2021), and the Forth Wave from Delta or B.1.617.2 variant (6 July to 31 December 2021) [7].

As of 1 January 2022, Thailand had reported 2,226,446 cumulative cases and 21,708 deaths [7]. Similar to many countries, Thailand implemented COVID-19 vaccination with two primary vaccines: Sinovac and AstraZeneca, beginning 28 February 2021 [8]. Later, other vaccines (Sinopharm, Pfizer, and Moderna) were imported in mid-2021 [9]. There was no descriptive study on the COVID-19 situation and vaccination in Thailand in 2021. Additionally, the association between vaccination rates and case rates at the provincial level was questionable. Thus, this study aimed to describe the characteristics of COVID-19 cases and vaccination recipients in Thailand in 2021. It determined if there was an association at the provincial level between a reduction in COVID-19 case rates and vaccination rates and related variables.

## 2. Materials and Methods

We conducted a secondary data analysis using two databases that reported during January–December 2021: Thai COVID-19 cases reported to the Department of Disease Control (DDC), Ministry of Public Health (MoPH); and Thai COVID-19 vaccination records reported to the MoPH Immunization Center, Office of the Permanent Secretary, MoPH.

### 2.1. Data Collection

Data were collected in Excel and aggregated into reports in the DDC database from 1 January 2021 to 31 December 2021 for COVID-19 cases. The DDC database defined a COVID-19 case as any person who tested positive for SARS-CoV-2 by the reverse transcription polymerase chain reaction (RT-PCR) test and reported it to the database. This database did not include cases detected by the antigen test kit. For vaccination reports, we included reports from the MoPH Immunization Center from 28 February 2021 to 31 December 2021 because the vaccinations began on 28 February 2021. We excluded imported COVID-19 patients reported in state quarantine (SQ) or alternative state quarantine (ASQ). Incomplete records with missing variables were excluded. To estimate an association between case rates and vaccination rates, we analyzed data at the provincial level and selected data since 1 April 2021 for analyses because vaccines were allocated to all provinces after that time.

For COVID-19 cases, the aggregated report included date, gender, age, patient type, and the province of treatment. For the aggregated report of COVID-19 vaccinations, it included vaccination date, gender, age, type of vaccine group, vaccination province, and dose.

Potential confounders at the ecological level (e.g., COVID-19 zones, curfew measures set by provincial authorities, the proportion of foreign workers, and the proportion of tourists) were included. COVID-19 zones were coded by color: dark red as maximum, strictly controlled areas; light red as maximum controlled areas; orange as controlled areas; yellow as high-level areas; green as surveillance areas; and blue as tourism areas. This information was available from the website for COVID-19 created by the Ministry of Interior (www.moicovid.com (accessed on 1 January 2022)). The proportion of foreign workers reporting each month was extracted from the statistics of foreign workers available at https://www.doe.go.th/ (accessed on 1 January 2022). The proportion of tourists in each province reporting each month was published at https://www.mots.go.th/ (accessed on 1 January 2022).

### 2.2. Data Analysis

Analyses were divided into two parts: characteristics and distribution of COVID-19 cases and vaccinations in Thailand between 1 January 2021 and 31 December 2021, grouped by gender; and age calculated and presented by percentage, median, and interquartile range. New daily COVID-19 cases and cumulative vaccination rates were mapped by month; we selected the period after 1 April 2021 (week 14) for spatial panel model analyses because all provinces in Thailand received vaccines after that time.

A spatial panel model analysis was used to determine an association between the COVID-19 case rates per 100,000 people and vaccination rates in each province, considering time lags at two, four, six, and eight weeks after vaccination.

A Chi-square test with a Poisson distribution was used to test spatial heterogeneity. Global Moran’s I with a Poisson distribution was used to test spatial dependence with a statistical significance cut-off level of 0.05.

We divided the estimation into three periods: overall (weeks 14–52), ascending phase (weeks 14–32), and descending phase (weeks 34–52). The units of analysis were at the provincial level. Bivariate analyses were conducted to explore an association between case rates and each variable. For multivariate analyses, we focused on the association between case rates and each variable over two weeks in three periods. A statistical significance cutoff level of 0.05 was applied. R Studio version 1.4.1 with the spml, spdep, lme4, and lmerTest packages was used to conduct these analyses.

### 2.3. Ethical Considerations

The Ethical Committee approved the Department of Disease Control study, Ministry of Public Health, Thailand (Ref. No. 64060 and Date of Approval: 23 August 2021).

## 3. Results

### 3.1. COVID-19 Cases and Case Rates, Per 100,000 People

In 2021, there were 1,965,023 confirmed cases by RT-PCR; the highest daily cases were reported on 13 August 2021 (21,770 cases) (Figure 1a). The median of daily reported cases was 3173.50 (IQR 8606.00–759.80). Most (1,724,092; 87.74%) were in the Fourth Wave of Thailand (5 July–31 December 2021, or weeks 28–53). Most were in the age groups 16–30 years old (29.60%) and 31–45 years old (29.20%). Among these, 51.80% were female (Table 1).

In early 2021, each province had less than 101 cases per 100,000 people until April, when cases were clustered in the central part of Thailand, Bangkok, and metropolitan areas. After April 2021, cases were distributed from the central to surrounding areas and reached a peak in August. Then, case rates in most of Thailand decreased, except in the southern part, which reported high case rates until November (Figure 1a and Figure 2a).

### 3.2. COVID-19 Vaccination Counts and Rates

The first COVID-19 vaccine in Thailand was administered on 28 February 2021. By the end of 2021, Thailand had administered 45,788,315 total cumulative first vaccination doses, accounting for 63.60% of the population. In Thailand, 57.70% and 8.74% of people received complete vaccinations and booster doses, respectively (Figure 1b). At the peak of COVID-19 new daily cases (13 August 2021), the coverage of the first dose and second dose was 22.59% and 0.82%, respectively. In fact, 26.90% of people who received the first dose were 31–45 years old. More females (52.20%) received the first dose compared to males (Table 1).

Vaccinations had been localized in the central part of Thailand by the end of February 2021. Vaccinations were then expanded to other regions, especially in Thailand’s northern, eastern, western, and southern regions (Figure 1b and Figure 2b).

### 3.3. COVID-19 Case Rates and Vaccination Rates

Spatial heterogeneity with a Poisson distribution among COVID-19 cases (per 100,000 people) and vaccination rates were determined for 1,069,759 (*p* < 0.01) and 39,273.54 (*p* < 0.01), respectively. Global Moran’s I with a Poisson distribution was tested among COVID-19 case rates per 100,000 people, and vaccination rates were shown for spatial dependence as 0.33 (*p* < 0.01) and 0.12 (*p* = 0.05), respectively.

The results of bivariate analyses in the overall period found that vaccination rates (week 0, lagged 2 wks, lagged 4 wks), color zones (yellow zone in lagged 4, 6, and 8 wks, orange zone in lagged 6 and 8 wks, light red zone in lagged 2, 4, 6, and 8 wks, and dark red zone in all times), curfew set by provincial authority, and proportion of migrants had positive relationships with case rates (*p* < 0.05). Meanwhile, a higher proportion of tourists had a negative relationship with case rates (*p* < 0.05) (Table 2).

Focusing on the ascending phase, we found that vaccination rates, color zones (yellow zone in lagged 4 and 6 wks, orange zone in lagged 8 wks, light red zone in all times except week 0, dark red zones in all times, and light red zone in all times except week 0), curfew set by provincial authorities, and proportion of migrants had a positive relationship with case rates (*p* < 0.01). Meanwhile, the proportion of tourists had a negative relationship with case rates (*p* < 0.01). Conversely, the descending phase found that vaccination rates, color zones (dark red zone in lagged 8 wks), curfew set by provincial authorities (lagged 8 wks), and proportion of tourists had negative relationships to case rates (*p* < 0.05). The color zone (dark red zone at all times except lagged 8 wks) and proportion of migrants had positive relationships with case rates (*p* < 0.01) (Table 2).

We excluded curfews set by provincial authorities from the multivariate analyses because they were part of color zone measures. Lagged two weeks of all variables were included. The overall period in the multivariate analysis found that the dark red zone and proportion of migrants had positive relationships with case rates (*p* < 0.01); meanwhile, the proportion of tourists had a negative relationship with case rates (*p* < 0.01). For the ascending phase, vaccination rate, the dark red zone, and proportion of migrants had positive relationships with case rates (*p* < 0.01). However, the yellow zone and the orange zone had negative relationships (*p* < 0.01). For the descending phase, vaccination rate and the light red zone had negative relationships with case rates (*p* < 0.01); however, the proportion of migrants had a positive relationship with case rates (*p* < 0.01) (Table 3).

## 4. Discussion

Since late 2019, the COVID-19 pandemic has emerged and is still ongoing. Because of the mutation of the SARS-CoV-2 virus, emerging variants contribute to the continuation of the pandemic. The World Health Organization (WHO) announced and updated the list of Variants of Interest (VOIs) and Variants of Concern (VOCs) to monitor and assess the evolution of SARS-CoV-2 [10,11]. In 2021, COVID-19 cases in Thailand were affected by three variants of SARS-CoV-2: B.1.36.16, or Bangladesh/UK lineage, Alpha, and Delta, which caused 1,965,023 cases. Most of the cases in 2021 (87.74%) were reported in the Delta variant period, in which this variant is more transmissible than the Alpha variant (40–60%) and almost twice as transmissible as the original strain [12]. Thus, the number of cases in 2021 was much higher than the cases in 2020 (6884 cases) affected by the original Wuhan strain [13]. Monitoring COVID-19 variants is helpful to understand the infectivity of each variant and control outbreaks based on the evidence [14,15]. In Thailand, the Department of Medical Sciences (MoPH) and 15 Regional Medical Sciences center (RMSc) laboratories started SARS-CoV-2 variant surveillance on 1 April 2021, which contributed to the understanding of the COVID-19 situation in Thailand and helped the MOPH plan strategies to control the spread [16].

By the end of 2021, 63.6% and 42.3% of people in Thailand will have received the first vaccination and the complete vaccination, respectively. It will have achieved the goal of the WHO target, which was to reach the 40% COVID-19 full vaccination population coverage target by the end of December 2021 [17]. The peak of COVID-19 new daily cases was on 13 August 2021, during the Delta variant predominance, in which the coverage of the first and second doses was 22.59% and 0.82%, respectively. Due to the small number of COVID-19 vaccines in the early period, the MoPH had to prioritize allocating vaccines to high-risk groups and high-spot areas [18]. So, the vaccination rates in high-case areas were higher than in low-case areas in the early period.

For the association between case rates and interested variables, we found that the vaccination rates in the ascending phase had a positive association. Conversely, there was a negative relationship in the descending phase. Thus, the overall period found that there was a slightly positive association. As discussed previously, vaccines were allocated to the hot spot areas in the early period, so it would not reflect the impact of vaccination. The vaccine can prevent severe COVID-19, especially in high-risk groups [19]. However, the effect of vaccination rates might not be fully explained in the descending phase. The decline in COVID-19 new daily cases may have occurred because of the dynamic of the COVID-19 infection. New cases would decrease after the peak incidence date due to herd immunity for the Alpha variant period [20]. Counterfactual scenarios should be studied further.

Color zones, especially the high restriction or dark red zone, had a positive association with case rates in each province in the overall period of multivariable analyses, which included only a two-week lag for interested variables. The restriction measure was implemented following a high incidence to prevent the spread, so the effect may not reflect immediately. Long lag times seemed to be negatively associated with case rates in the bivariable analyses. However, a study found a strong relationship between the implementation of social distancing policies resulting in decreased mobility and reduced COVID-19 case growth two or three weeks later [21]. The performance of color zone measures should be evaluated.

Another variable that had a positive association with case rates in each province was the proportion of migrant workers. The cause of the second wave in Thailand was the B.1.36.16 or Bangladesh/UK lineage, in which undocumented migrant workers were suspected to be the origin of the Samut Sakhon cluster [22]. Undocumented migrant workers were allowed to register for a work permit during the pandemic [23]. However, some undocumented migrant workers might not have registered. Thus, the Center for COVID-19 Situation Administration (CCSA) directed all provinces in Thailand to conduct proactive screening of migrant workers who entered illegally [24,25]. Migrants, regardless of work permit document status, should receive a vaccination similar to other people in Thailand, which would produce the most effective outcome in terms of health and economy [26]. Additionally, active case-finding among migrants would be combined to reduce COVID-19 morbidity and mortality [27].

The proportion of tourists had a negative relationship with case rates because tourists may perceive the risk of getting an infection and not take the chance to travel [28]. COVID-19 has impacted the tourism industry, especially in Thailand, with high proportions of income from this industry. Multidisciplines, such as tourism and public health, should collaborate, prepare, and plan to recover in a new manner during or after the pandemic [29].

This study had several limitations. First, we cannot explore the vaccination history among cases at an individual level because these two databases did not link, and this variable was not complete in the COVID-19 database. The ecological fallacy was also an issue in the interpretation of the results. Second, cases in this database were only from PCR-confirmed tests and might be lower than the actual situation. However, this study still reveals insight into the situation of COVID-19 in Thailand. Third, ecological-level confounders, such as color zone measures, curfews at the provincial level, migrants, and tourists, were considered and included in the analyses. However, there might be other ecological-level confounders that we cannot control in the analyses. Last, the reverse causality between COVID-19 vaccination rates and case rates was established in this study. Otherwise, we tried to solve this issue by dividing the analyses’ phases into ascending and descending phases.

## 5. Conclusions

In Thailand, 31–45 years old was the age group with the highest proportion of cases and the highest vaccination rate (29.20% and 26.90%, respectively). Vaccines were distributed to hot-spot pandemic areas early, concurring with high case rates. Overall, vaccination rates had a slightly positive association with case rates. Focusing on specific periods, vaccination rates had a positive association in the ascending phase and a negative association in the descending phase. The proportion of migrant workers and the color zones had positive associations with case rates. Meanwhile, the proportion of tourists had a negative association with case rates. It may be because of the perception that infections occur during travel. Vaccination and active case-finding should be targeted at migrants. Thus, the tourism and public health industries should collaborate, prepare, and plan for the future of the tourism industry.

## Figures and Tables

**Figure 1 tropicalmed-08-00311-f001:**
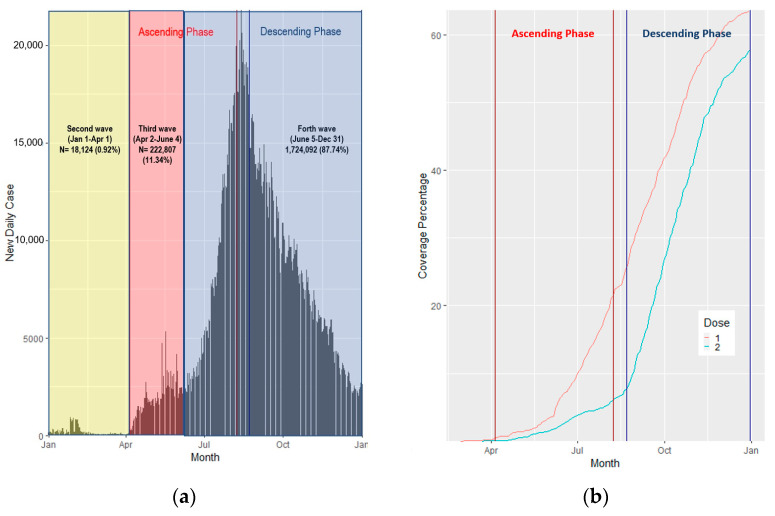
(**a**) COVID-19 new daily cases in Thailand by month; (**b**) COVID-19 coverage percentage by month and dose.

**Figure 2 tropicalmed-08-00311-f002:**
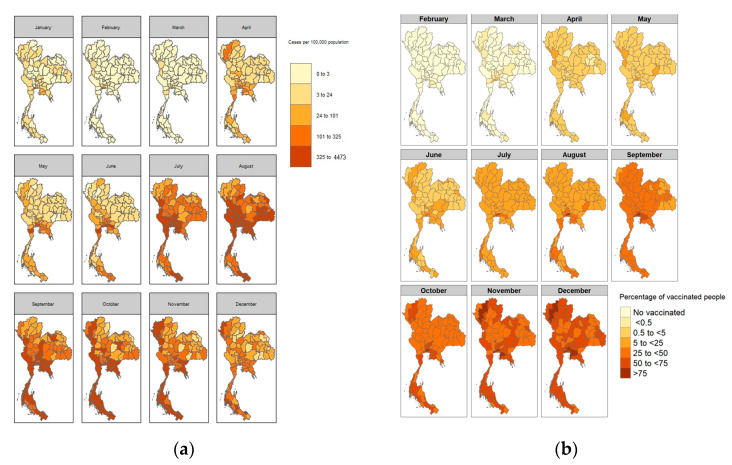
(**a**) COVID-19 case rates per 100,000 people by month; (**b**) COVID-19 first dose coverage percentage by month.

**Table 1 tropicalmed-08-00311-t001:** Characteristics of COVID-19 cases and those vaccinated.

Characteristics	Cases (%)	Vaccinated (%)
Age; Median (IQR)	34 (47–23)	
- Less than 15 years old	266,055 (13.5)	2,204,217 (4.8)
- 16 to 30 years old	581,236 (29.6)	11,675,207 (25.5)
- 31 to 45 years old	574,452 (29.2)	12,318,571 (26.9)
- 46 to 60 years old	363,541 (18.5)	11,519,634 (15.2)
- 61 to 75 years old	140,066 (7.1)	6,330,258 (13.8)
- 76 to 90 years old	36,959 (1.9)	1,658,310 (3.6)
- More than 90 years old	2714 (0.1)	82,118 (0.2)
Gender		
- Male	946,355 (48.2)	21,888,301 (47.8)
- Female	1,018,668 (51.8)	23,900,014 (52.2)
Total	1,965,023	45,788,315 ^1^

^1^ Vaccination of at least one dose.

**Table 2 tropicalmed-08-00311-t002:** Bivariate analyses of COVID-19 case rates and related variables.

Variables	Time	Overall		Ascending Phase (wk 14–32)		Descending Phase (wk 34–52)	
		Estimate (95% CI)	*p*-Value	Estimate (95% CI)	*p*-Value	Estimate (95% CI)	*p*-Value
Vaccinated Rates	Week 0	0.14 (0.07, 0.20)	<0.001	2.15 (1.89, 2.42)	<0.01	−0.52 (−0.67, −0.36)	<0.001
	Lagged 2 wks	0.11 (0.04, 0.17)	<0.001	2.42 (2.11, 2.74)	<0.01	−0.59 (−0.73, −0.45)	<0.001
	Lagged 4 wks	0.08 (0.01, 0.15)	0.02	2.68 (2.30, 3.06)	<0.01	−0.65 (−0.79, −0.51)	<0.001
	Lagged 6 wks	0.06 (−0.02, 0.13)	0.14	3.14 (2.64, 3.64)	<0.01	−0.73 (−0.88, −0.59)	<0.001
	Lagged 8 wks	0.03 (−0.05, 0.10)	0.50	3.96 (3.28, 4.63)	<0.01	−0.83 (−0.97, −0.68)	<0.001
COVID-19 Zones	Week 0						
	- Green zone	-	-	-	-	-	-
	- Yellow zone	−0.63 (−13.42, 12.16)	0.92	−2.47 (−12.99, 8.06)	0.65	-	-
	- Orange zone	−2.37 (−14.40, 9.66)	0.70	−4.79 (−14.39, 4.81)	0.33	−7.35 (−17.28, 2.58)	0.25
	- Light red zone	10.51 (−1.46, 22.48)	0.09	10.01 (0.17, 19.83)	0.05	2.74 (−6.41, 11.89)	0.56
	- Dark red zone	71.74 (59.03, 84.45)	<0.001	93.16 (81.19, 105.12)	<0.01	49.95 (39.15, 60.74)	<0.001
	- Blue zone	24.92 (5.56, 44.29)	0.01	-	-	−6.54 (−25.75, 12.66)	0.50
	Lagged 2 wks						
	- Green zone	-	-	-	-	-	-
	- Yellow zone	0.90 (−7.44, 9.23)	0.83	−0.14 (−6.79, 6.51)	0.97	-	-
	- Orange zone	−2.68 (−9.75, 4.39)	0.46	−5.13 (−10.67, 0.40)	0.07	−3.60 (−15.82, 8.62)	0.56
	- Light red zone	7.20 (0.28, 14.11)	0.04	9.12 (2.98, 15.25)	<0.01	−2.74 (−14.13, 8.66)	0.64
	- Dark red zone	48.08 (40.08, 56.07)	<0.001	83.82 (72.70, 94.95)	<0.01	34.12 (21.34, 46.89)	<0.001
	- Blue zone	14.24 (−4.43, 32.92)	0.13	-	-	−8.12 (−29.84, 13.61)	0.46
	Lagged 4 wks						
	- Green zone	2.84 (−5.54, 11.22)	0.51	1.67 (−4.61, 7.95)	0.60	-	-
	- Yellow zone	8.50 (0.75, 16.25)	0.03	6.18 (0.27, 12.08)	0.04	-	-
	- Orange zone	2.79 (−3.75, 9.32)	0.40	−1.11 (−6.22, 4.01)	0.67	−11.46 (−33.27, 10.36)	0.30
	- Light red zone	8.82 (2.46, 15.18)	0.01	10.25 (3.98, 16.52)	<0.001	−12.82 (−24.57, 8.93)	0.25
	- Dark red zone	34.03 (26.88, 41.18)	<0.001	74.40 (62.41, 86.51)	<0.001	9.73 (−12.69, 32.15)	0.40
	- Blue zone	11.30 (−11.16, 33.75)	0.32	-	-	−23.34 (−54.22, 7.55)	0.14
	Lagged 6 wks						
	- Green zone	2.04 (−5.18, 9.27)	0.58	0.87 (−4.68, 6.41)	0.76	-	-
	- Yellow zone	10.52 (3.96, 17.09)	0.01	8.28 (2.43, 14.14)	<0.001	-	-
	- Orange zone	5.80 (0.72, 10.89)	0.03	1.39 (−2.81, 5.59)	0.52	−1.42 (−1.88, 15.15)	0.87
	- Light red zone	10.08 (5.20, 14.96)	<0.001	10.87 (5.09,16.64)	<0.001	−0.60 (−16.74, 15.54)	0.94
	- Dark red zone	22.66 (16.84, 28.49)	<0.001	65.53 (51.48, 79.58)	<0.001	3.12 (−14.34, 20.59)	0.73
	- Blue zone	3.42 (−26.36, 33.21)	0.82	-	-	−26.60 (−59.25, 6.05)	0.11
	Lagged 8 wks						
	- Green zone	2.39 (−4.52, 9.30)	0.50	1.43 (−4.23, 7.08)	0.62	-	-
	- Yellow zone	9.92 (3.66, 16.19)	0.01	1.63 (−11.02, 14.27)	0.80	-	-
	- Orange zone	10.30 (5.68, 14.93)	<0.001	4.21 (0.22, 8.20)	0.04	9.91 (−0.15, 19.97)	0.05
	- Light red zone	13.76 (9.34, 18.18)	<0.001	16.17 (10.35, 22.00)	<0.001	1.44 (−6.11, 9.01)	0.71
	- Dark red zone	12.54 (7.11, 17.98)	<0.001	58.22 (41.46, 74.97)	<0.001	−19.02 (−29.16, −8.88)	<0.001
	- Blue zone	−4.02 (−54.23, 46.19)	0.88	-	-	−31.66 (−79.55, 16.24)	0.20
Curfew	Week 0	56.43 (51.36, 61.49)	<0.001	66.48 (59.04, 73.92)	<0.001	49.43 (42.65, 56.22)	<0.001
	Lagged 2 wks	40.97 (36.04, 45.91)	<0.001	61.11 (51.73, 70.49)	<0.001	37.21 (30.25, 44.16)	<0.001
	Lagged 4 wks	27.63 (22.80, 32.47)	<0.001	56.44 (46.00, 66.89)	<0.001	22.48 (14.99, 29.97)	<0.001
	Lagged 6 wks	15.96 (11.20, 20.72)	<0.001	39.45 (27.64, 51.26)	<0.001	4.98 (−2.92, 12.89)	0.21
	Lagged 8 wks	5.57 (0.80, 10.34)	0.02	30.59 (17.6, 43.53)	<0.001	−20.68 (−29.27, −12.09)	<0.001
Migrants	Week 0	7.46 (5.74, 9.17)	<0.001	8.26 (6.40, 10.13)	<0.001	7.17 (4.93, 9.40)	<0.001
	Lagged 2 wks	7.80 (6.07, 9.53)	<0.001	8.46 (6.59, 10.33)	<0.001	7.09 (4.86, 9.32)	<0.001
	Lagged 4 wks	8.17 (6.43, 9.91)	<0.001	8.55 (6.67, 10.43)	<0.001	7.15 (4.92, 9.37)	<0.001
	Lagged 6 wks	8.38 (6.63, 10.13)	<0.001	8.63 (6.74, 10.51)	<0.001	7.03 (4.81, 9.25)	<0.001
	Lagged 8 wks	8.33 (6.60, 10.07)	<0.001	8.65 (6.76, 10.55)	<0.001	6.79 (4.58, 9.00)	<0.001
Tourists	Week 0	−0.10 (−0.20, −0.01)	0.03	−0.39 (−0.66, −0.13)	<0.001	−0.43 (−0.56, −0.29)	<0.001
	Lagged 2 wks	−0.23 (−0.34, −0.13)	<0.001	−0.30 (−0.49, −0.12)	<0.001	−0.49 (−0.63, −0.35)	<0.001
	Lagged 4 wks	−0.39 (−0.51, −0.27)	<0.001	−0.24 (−0.38, −0.09)	<0.001	−0.60 (−0.77, −0.43)	<0.001
	Lagged 6 wks	−0.54 (−0.67, −0.42)	<0.001	−0.26 (−0.39, −0.13)	<0.001	−0.79 (−1.00, −0.57)	<0.001
	Lagged 8 wks	−0.60 (−0.72, −0.47)	<0.001	−0.32 (−0.46, −0.19)	<0.001	−0.93 (−1.20, −0.65)	<0.001

**Table 3 tropicalmed-08-00311-t003:** Multivariate analyses of COVID-19 case rates and related variables with a lag of two weeks.

Variables	Overall		Ascending Phase (wk 14–32)		Descending Phase (wk 34–52)	
	Estimate (95% CI)	*p*-Value	Estimate (95% CI)	*p*-Value	Estimate (95% CI)	*p*-Value
Vaccinated Rates (Lagged 2 wks]	0.05 (−0.07, 0.17)	0.40	1.94 (1.54, 2.34)	<0.001	−0.80 (−1.09, −0.52)	<0.001
COVID-19 zones (Lagged 2 wks]						
- Green zone	-	-	-	-	-	-
- Yellow zone	−1.69 (−10.63, 7.24)	0.71	−10.71 (−18.55, −2.88)	<0.001	-	-
- Orange zone	−5.92 (−13.67, 1.84)	0.13	−9.74 (−16.27, −3.21)	<0.001	−11.32 (−25.37, 2.74)	0.11
- Light red zone	3.33 (−5.09, 11.76)	0.44	1.41 (−6.31, 9.13)	0.72	−18.62 (−33.41, −3.82)	0.01
- Dark red zone	42.63 (32.12, 53.15)	<0.001	64.18 (50.70, 77.66)	<0.001	17.56 (−0.56, 35.68)	0.06
- Blue zone	12.57 (−7.82, 32.97)	0.23	-	-	−10.70 (−34.08, 12.67)	0.37
Migrants (Lagged 2 wks]	6.44 (4.90, 7.99)	<0.001	3.12 (1.49, 4.76)	<0.001	8.38 (6.03, 10.72)	<0.001
Tourists (Lagged 2 wks]	−0.28 (−0.45, −0.10)	<0.001	0.06 (−0.19, 0.32)	0.62	0.14 (−0.18, 0.47)	0.39

## Data Availability

Not applicable.

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
