# Peer review of "Impact of COVID-19 Vaccination Rates and Public Measures on Case Rates at the Provincial Level, Thailand, 2021: Spatial Panel Model Analyses"

_tropicalmed, 2023, doi:10.3390/tropicalmed8060311_

Round 1

Reviewer 1 Report

Jitpeera and colleagues described the characteristics of COVID-19 cases and vaccination in Thailand in 2021. This paper requires major revision addressing the comments below:

1. Kindly update the epidemiologic data here for COVID-19 prevalence since the data reported were taken from January 1, 2022.

2. Kindly emphasize that in the early part of the pandemic the high influx of cases in the hospitals overwhelmed many diagnostic laboratories especially from the developing world and might have caused possible cross-contamination making COVID diagnosis even more challenging DOI: 10.21141/PJP.2020.09

3. In the introduction, kindly emphasize the novelty of this work. Most of the data gathered here were from 2021. What about 2022 onwards? Please expound on the relevance of this study since we are now approaching the post-COVID era.

4. This paper requires extensive english editing as there were many grammatical errors observed in the text.

5. Kindly discuss in depth the role of vaccination in preventing severe COVID-19 and being immunoprotective for people with compromised immunity. DOI: 10.3390/vaccines11040724

6. How did the emergence of variants of concerns impacted vaccination rate in Thailand? How could the results of this study help in framing nationwide policies to prevent future disease outbreaks?

This paper requires an extensive english editing.

Reviewer 2 Report

This study aimed to describe the characteristics of COVID-19 cases and vaccination people, Thailand, 2021. Likewise, it was determined if there was an association at the provincial level with a reduction in COVID-19 case rates, by vaccination and related variables.

This study has many limitations, as the authors point out. Considering that there were limitations that could not be overcome, the study is well planned, has clear objectives and adequate methodology. The presentation of the results is clear. My main criticism is in the discussion of results.

What are the implications of having found that vaccination rates had a positive association in the ascending phase and a negative association in the descending phase?

Authors should consult the scientific literature on the impact of vaccination in other populations to compare their results and interpret the situation in Thailand.

Minor corrections

Lines 260-262: Delete phrase: “In 2021, Thailand had 1,965,023 cumulative COVID-19 cases, of which most cases (87·74%) were reported in the Fourth Wave. Thailand has implemented COVID-19 vaccination since Feb 28, 2021. At the end of 2021, 63·60% (45,788,315) of the Thai population received one dose vaccination and 57·70% were fully vaccination”… This is not a conclusion based on the objective of the study.

Round 2

Reviewer 1 Report

The paper has been extensively revised and can now be accepted for publication.